# Molybdenum Disulfide Quantum Dots Prepared by Bipolar-Electrode Electrochemical Scissoring

**DOI:** 10.3390/nano9060906

**Published:** 2019-06-21

**Authors:** Yang Li, Xiaoxia Wang, Mengli Liu, Heng Luo, Lianwen Deng, Lei Huang, Shuang Wei, Congli Zhou, Yuanhong Xu

**Affiliations:** 1College of Materials Science and Engineering, Qingdao University, Qingdao 266071, China; liyang4875@outlook.com (Y.L.); wxx@qdu.edu.cn (X.W.); 15705420340@163.com (M.L.); 11181011018@stu.ouc.edu.cn (S.W.); 2017021070@qdu.edu.cn (C.Z.); 2College of Life Sciences, Qingdao University, Qingdao 266071, China; lei_hl@126.com; 3College of Physics and Electronics, Institute of Super-Microstructure and Ultrafast Process in Advanced Materials, Central South University, Changsha 410083, China; luohengcsu@csu.edu.cn (H.L.); denglw@csu.edu.cn (L.D.)

**Keywords:** molybdenum disulfide, bipolar-electrode, electrochemical method, quantum dots, electromagnetic wave absorption

## Abstract

A convenient bipolar-electrode (BPE) electrochemical method was engineered to produce molybdenum disulfide (MoS_2_) quantum dots (QDs) using pure phosphate buffer (PBS) as the electrolyte and the MoS_2_ powder as the precursor. Meanwhile, the corresponding by-product precipitate was studied, in which MoS_2_ nanosheets were observed. The BPE design would not be restricted by the shape and size of the MoS_2_ precursor. It could lead to the defect generation and 2H → 1T phase variation of the MoS_2_, resulting in the formation of nanosheets and finally the QDs. The as-prepared MoS_2_ QDs exhibited high photoluminescence (PL) quantum yield of 13.9% and average lateral size of 4.4 ± 0.2 nm, respectively. Their excellent PL property, low cytotoxicity, and good aqueous dispersion offer promising applicability in PL staining and cell imaging. Meanwhile, the as-obtained byproduct containing the nanosheets could be used as an effective electromagnetic wave (EMW) absorber. The minimum reflection loss (RL) value was −54.13 dB at the thickness of 3.3 mm. The corresponding bandwidth with efficient attenuation (<−10 dB) was up to 7.04 GHz (8.8–15.84 GHz). The as-obtained EMW performance was far superior over most previously reported MoS_2_-based nanomaterials.

## 1. Introduction

MoS_2_ belongs to one two-dimensional (2D) group-VIB transition metal dichalcogenides (TMDs), which has gained significant attention due to their large surface area (SSA), distinct electrical properties and tunable bandgaps [1]. The MoS_2_ nanomaterials have shown great application potentials in catalysis [2,3,4], electromagnetic wave (EMW) adsorption [5,6], biomedical applications [6], etc. For example, MoS_2_ nanosheets were confirmed to be a promising EMW absorber not only because of their high SSA, but also due to the defect dipole polarization resulting from Mo and S vacancies as well as the phase transition between the trigonal prismatic (2H) semiconducting phase to octahedral (1T) metallic one [7,8,9,10]. It has an indirect band gap of 1.2 eV in its bulk form, while becomes 1.9 eV when thinned to the monolayer nanosheets, indicating the formation of a direct band-gap semiconductor [2,11,12]. Furthermore, PL could emerge when the thickness of MoS_2_ crystals became smaller [12]. Besides the layer number, the nanoscale size of the MoS_2_ would also contribute to the PL emission due to the edge effects and quantum confinement [2]. As a result, PL MoS_2_ quantum dots (QDs) could be obtained. Many approaches have been proposed to synthesize the MoS_2_ QDs including the electro-Fenton processing [8], Li-intercalation [13,14], ultrasonication [15,16], hydrothermal/solvothermal treatment, and CVD approaches [17]. Despite the achievements, they are still restricted by the time consumption, or poor environmental tolerance, or expensive instrument or hazardous organic solvents [8]. Therefore, there is a great need to discover a simple, efficient, non-toxic, and cost-effective method for preparing MoS_2_ nanomaterials.

The electrochemical method has been widely used in exfoliating the 2D materials due to its superiorities including simple operation, low consumption, good reproducibility [8], etc. Applications of the traditional electrochemical methods in preparing MoS_2_ QDs have also been explored [17,18]. Therein, precursor in bulky form have to be applied as the working electrodes (or anodes). However, the MoS_2_ precursor popularly exists in the form of powder, which greatly limits its wide application in the preparation of MoS_2_ QDs.

Bipolar-electrode (BPE) method is a special type of electrochemical method that has attracted much attention recently. Its device is mainly connected to a driving power source (such as the Pt electrode, Au electrode, etc.) and a conductor immersed in the electrolyte (i.e., bipolar electrode, BPE) composition. A BPE can act as an anode and a cathode at the same time, on which the oxidation and reduction reactions occur at the respective end. This method has the advantages of simplicity, easy operation, and especially not limited by the shape of the precursor conductor [18]. Accordingly, large-size conductive 2D materials such as black phosphorus [19] and WS_2_ [20] can be electrolyzed into nano-sized particles, despite the precursor is in bulky form or powder. However, as far as we know, there is no report about the preparation of the QDs derived from the 2D materials via the BPE methods, not to mention the MoS_2_ QDs. In addition, upon the BPE treatment for the 2D materials, only the nano-sized particles were paid close attention, while the by-product precipitate has been neglected, which might provide useful information for elucidating the BPE process and hold great application potential.

In this study, the BPE electrochemical strategy was applied for preparing MoS_2_ QDs using the MoS_2_ powder and PBS as the precursor and electrolyte, respectively. Meanwhile, the corresponding by-product precipitate was also studied, in which the MoS_2_ nanosheets were obtained. The electrochemical process was not restricted with the shape of the MoS_2_ precursor. It showed the advantages of simplicity and environmental friendliness. The as-prepared MoS_2_ QDs exhibit good PL, high crystallinity, good dispersion, and narrow particle size distribution. Their advantages offer promising applicability in PL staining and cell imaging. The as-obtained precipitate containing the nanosheets was also tested for the EMW absorption. Through the various characterizations, we proposed the generation mechanism of the MoS_2_ QDs and nanosheets via the BPE electrochemical strategy.

## 2. Materials and Methods

Reagents: Bulk MoS_2_ powder (mol wt 160.07, purity 98.0%), sodium dihydrogen phosphate dehydrate (mol wt 156.01, purity 99.0%), disodium hydrogen phosphate dodecahydrate (mol wt 358.14, purity 99.0%), sodium hydroxide (granular, mol wt 40, purity 96.0%), ethanol (mol wt 46.07, purity 99.7%), and potassium bromide (mol wt 119, purity 99.0%) was supplied by Sinopharm Chemical Reagent Co., Ltd., Shanghai, China. Ammonium fluoride (NH_4_F, mol wt 37.04, purity ≥ 96%), concentrated sulfuric acid (H_2_SO_4_, mol wt 98.04, density 1.84 g cm^−3^). 3-(4,5-dimethyl-2-thiazolyl)-2,5-diphenyltetrazolium bromide (MTT, mol wt 414.32, purity 98%), lysogeny broth (LB) medium, Dulbecco’s modified Eagle medium (DMEM). All other reagents were of analytical grade and applied as received. Deionized distilled water was applied in the experiment.

Apparatus: The RXN-305D DC power supply was obtained from Shenzhen Zhaoxin Yuan Electronics Co., Ltd. (Shenzhen, China). The LCD numerical controlled heating type magnetic stirrer, model MS-H-Pro+, is provided by Dragon Laboratory Limited (Beijing, China). The high-speed table centrifuge model of TGL-15B was provided by Shanghai anting scientific instrument factory. The morphology of MoS_2_ QDs was observed on a JEOL Ltd. JEM-2010 transmission electron microscope (JEOL Ltd., Beijing, China) and atomic force microscopy using a SPI3800N microscope operating in the tapping mode (Seiko Instruments Inc., Shenzhen, China). The XRD was carried out on a Rigaku D-MAX 2500/PC with the Cu K α radiation (λ = 1.54056 Å) (Tokyo, Japan). FTIR was carried out on a Nicolet 5700 FTIR spectrometer (Thermo Electron Scientific Instruments Corp., Shanghai, China). The Raman spectra were performed using a DXR2 micro Raman imaging spectrometer (Thermo Fisher, Waltham, MA, U.S). XPS data were collected on an ESCALab220i-XL electron spectrometer (VG Scientific, West Sussex, U.K.) using 300 W Al Kα radiation. The FL spectra were performed on an Edinburgh instruments spectrofluorometer FS5 (Edinburgh, U.K.) with the excitation/emission slits of 5.0 × 5.0 nm. The ultraviolet–visible (UV–vis) absorbance was gained from a Mapada UV-6300 double beam spectrophotometer (Shanghai, China). The electromagnetic parameters were evaluated by an Agilent AV3618 vector network analyzer.

Preparation of MoS_2_ QDs and precipitate containing the nanosheets: The MoS_2_ QDs and precipitate containing the nanosheets were synthesized by a low cytotoxicity, simple, and nondestructive BPE electrochemical system. Typically, PBS (0.2 M, pH = 7.4) was used as the electrolyte solution, and MoS_2_ powder (0.173 g) was placed in a beaker (50 mL). Two platinum (Pt) sheets (area > 1.0 cm^2^) were used as the anode and the cathode, respectively. Different electrolyte compositions all at 0.2 M (PBS, NH_4_F, and H_2_SO_4_), applied voltages (3 V, 5 V, and 7 V) and reaction times (0–30 h) were optimized for the preparation. The electrochemical exfoliation was conducted accompanying with stirring at 500 rpm for the whole experiment. The above-mentioned reaction solution was under centrifugation at 12,000 rpm for 15 min to reach a colorless dispersion containing MoS_2_ QDs and grey-black precipitation, respectively. The colorless dispersion was dialyzed in a 1000 Da dialysis bag against deionized water for 12 h to remove phosphate and obtain purified MoS_2_ QDs. The grey-black precipitates are dried by freeze-drying to for further characterization and applications.

## 3. Results and Discussion

### 3.1. Preparation and Characterization of the MoS_2_ QDs and Precipitate Containing the Nanosheets

As shown in Scheme 1, MoS_2_ QDs and nanosheets were prepared through a simple BPE electrochemical system using MoS_2_ powder as the precursor. The products were finally centrifuged to collect the supernatant and precipitate, respectively. Different applied voltages and reaction times electrolyte compositions were tested for the preparation. Bubbles were observed on the platinum electrodes during electrolysis, indicating the possible evolution of oxygen and hydrogen due to the electrolysis of water. As shown in Appendix A, the MoS_2_ QDs can be obtained at different experimental conditions, indicating the universality of the BPE synthesis strategy. Furthermore, the PL quantum yield of the as-prepared MoS_2_ QDs was evaluated based on the method indicated in the Supporting Information [21,22,23,24,25,26]. However, too low applied voltage (e.g., 3 V) would result in too low of an exfoliation efficiency, thus resulting in too long of a process. While too high voltage (e.g., 7 V) would lead to high current in the BPE system. In addition, bubbles generation on the platinum electrodes would be accelerated at higher applied voltage. Both the higher current and more generated bubbles would cause faster loss of the water, leading to unstable composition of the reaction system. Accordingly, 5 V was chosen as the optimized applied voltage. Meanwhile, it was found that the PL quantum yield of the MoS_2_ QDs increased with the increasing electrolysis time from 0 to 20 h, while longer time than 20 h would not only offer little positive effect on the quantum yield but also lead to great loss of the water. Thus, 20 h was selected as the optimum reaction time. Meanwhile, the PBS (pH = 7.4) showed the highest PL quantum yield among the tested electrolytes under the same applied voltage and reaction time. As a result, the optimum conditions were set as follows: applied voltage: 5 V, reaction time: 20 h and electrolyte: 0.2 M PBS (pH 7.4), which was used for the further experiments. The highest PL quantum yield of the as-prepared MoS_2_ QDs was calculated to be 13.9%.

The optical features of the as-prepared MoS_2_ QDs (0.2 M PBS, 5 V, 20 h) were studied by the UV–vis absorption and the PL spectroscopy accordingly. As shown, distinct absorption peaks at 221 nm was observed in the UV–vis absorption (Figure 1A), which should be ascribed to the excitonic characteristics of the MoS_2_ QDs [21,22,23]. Meanwhile, the 1931 CIE chromaticity diagram of the MoS_2_ QDs (Figure 1B) shows that the PL was in the blue emission region (0.14, 0.05) upon the excitation wavelength at 310 nm, which well matched with the UV–vis adsorption result. The PL of MoS_2_ QDs was collected upon different excitation wavelengths between 300 and 370 nm (Figure 1C). The emission spectra changed from 420 to 466 nm (2.66–2.95 eV) accordingly. As illustrated in Figure 1A, the maximum emission was obtained at 420 nm (2.95 eV) under the excitation wavelength at 310 nm (4 eV). The excitation-dependent PL emission is consistent with previous reports of the MoS_2_ QDs [8,24,25]. The PL decay curve of MoS_2_ QDs is shown in Figure 1D. The curve was fitted with a biexponential function, showing two dominant excitonic processes with nanosecond luminescence lifetime. Two excitonic lifetime of 1.25 ns (ca. 30.81%) and 5.57 ns (ca. 69.19%) can be obtained, respectively. In addition, the average PL lifetime of the as-prepared MoS_2_ QDs was estimated to be 4.24 ns.

In Figure 2A, the typical TEM image of the supernatant showed mono-dispersed nanodots, showing the possible formation of the MoS_2_ QDs with good dispersion in the aqueous medium. The nanodots were highly uniform with the average dimension of 4.4 ± 0.3 nm (Figure 2B). The illustration in Figure 2A is a high-resolution TEM (HRTEM) image of MoS_2_ QDs. The corresponding lattice spacing at 0.19 nm corresponds to the (100) plane, which was consistent with the reported values (JCPDS: 37-1492). The atomic force microscopy (AFM) image was also collected for the MoS_2_ QDs (Figure 2C). Besides the further confirmation of the good-dispersibility, the corresponding height analysis indicated that the thickness of the MoS_2_ QDs was between 1.95 and 2.0 nm and 1.98 ± 0.02 nm in average (the inset of Figure 2C), suggesting that the MoS_2_ QDs were highly exfoliated and in few layer [27,28,29].

To study the generation process of the MoS_2_ QDs, the TEM characterization were also carried out for the precipitate generated at electrolysis times of 5, 10, 15, and 20 h, respectively. As shown, compared with the MoS_2_ precursor form (Figure 2D), electrolysis time of 5 h (Figure 2E) does not affect the morphology of MoS_2_ sheets. With the reaction time extended to 10 h, MoS_2_ sheets with smaller thickness and smaller lateral dimension (95.8–231.6 nm) can be observed (Figure 2F). With the ongoing reaction to 15 h, much smaller nanosheets ranging from 11.2 nm to 61.4 nm appeared on the surface of the MoS_2_ sheets (Figure 2G). Subsequently, the nanosheets could be completely separated from the MoS_2_ sheets (Figure 2H) accompanying with the formation of nanodots on the independent nanosheets (Figure 2I) at 20 h. Meanwhile, independent nanodots—i.e., MoS_2_ QDs—could be reached since they were well dispersed in the aqueous supernatant (Figure 2A) [30].

XRD patterns of the MoS_2_ QDs and the precipitate generated at 20 h were then investigated while using that of the MoS_2_ precursor for comparison. The precursor shows an intensive diffraction peak at 2θ = 14.4° and four weaker peaks at 2θ = 29.026°, 32.68°, 39.54°, and 49.79°, respectively (curve a in Figure 3A). These signals were ascribed to the (002), (004), (100), (103), and (105) lattice planes of MoS_2_, respectively. Other weaker peaks are also characteristic peaks of MoS_2_ precursor. While for the precipitate (curve b in Figure 3A), all the other characteristic peaks of MoS_2_ were maintained except for the 29.026° (004) peaks. The disappearance of pristine (004) planes suggested the possible variation of MoS_2_ from 2H to metallic 1T phase [31]. The formation of a bulge between 16.09° to 31.5° should be due to the deterioration of the crystallization of the MoS_2_ precipitate [32]. While for the MoS_2_ QDs (curve c in Figure 3A), most of the other peaks were disappeared due to their highly exfoliated structure. Only two peaks could be observed at 2θ = 14.4° (002) and 32.68° (100), respectively. Furthermore, the obvious signal decrease of the (002) indicates the formation of MoS_2_ QDs in a few layers. The surface functional groups on MoS_2_ QDs and precipitate were studied through the Fourier transform infrared (FTIR) (Figure 3B). As shown, the MoS_2_ precursor and precipitate have similar absorption peaks, indicating their similar surface chemistry. While for MoS_2_ QDs, the peaks at 3449, 952, 898, and 465 cm^−1^ should be attributed to the stretching vibration of C–OH, Mo=O, S–OH, and Mo–S, respectively [22,24,33]. Also, the vibrational absorption of C=O at 1600 cm^−1^ could be observed [22]. However, the Mo–OH bending vibration at 1384 cm^−1^ and S–H at 619 cm^−1^ disappeared compared with the MoS_2_ precursor and the precipitate form [33]. It can obtain that the S–OH and Mo=O functional groups were generated on the MoS_2_ QDs surface, suggesting that the MoS_2_ QDs are slightly oxidized. Raman spectroscopy was further applied to understand the chemical bonding of the products. Generally, these peak position shifts in Raman spectroscopy can be used to recognize the layer thicknesses of the 2D layer material [34]. The Raman spectra of MoS_2_ precursor showed two distinct characteristic peaks being ascribed to the high energy in-plane vibration A_1g_ at 405.2 cm^−1^ and the lower energy out-of-plane vibration E^1^_2g_ at 378 cm^−1^ (curve a in Figure 3C) [35,36]. The position of the E^1^_2g_ and A_1g_ peak in the MoS_2_ precipitate (curve b in Figure 3C) shows that the intrinsic hexagonal lattices in MoS_2_ were still maintained. While the E^1^_2g_ and A_1g_ modes in the precipitates showed red and blue shifts, respectively, compared with those of MoS_2_ precursor. Accordingly, the position difference value between the two modes changed from 27.2 cm^−1^ to 25.9 cm^−1^. Accordingly, it is speculated that the MoS_2_ layered structure was destroyed during the electrolysis process, resulting in the reduced MoS_2_ layers, smaller sizes, and more defects, which were consistent with previously reported results [37,38]. Meanwhile, blue shifts of A_1g_ mode (curve c in Figure 3C), reduced frequency difference between E^1^_2g_ and A_1g_ modes (22.2 cm^−1^) and more abundant Raman peaks were observed for the MoS_2_ QDs than those for both the precursor and precipitate [37,38]. Therein, the abundant peaks should be attributed to the defects generated at the surfaces and edges of the MoS_2_ QDs, which are significant factor for the corresponding FL emission [37,39].

X-ray photoelectron spectroscopy (XPS) was further measured to study the elemental composition and surface state of the as-obtained MoS_2_ samples. As observed, the MoS_2_ precursor (above curve in Figure 3D) showed its 2H-phase based on the Mo 3d_5/2_ and Mo 3d_3/2_ orbitals at 229.3 eV and 232.4 eV, as well as the S 2p_3/2_ at 161.4 eV and the S 2p_1/2_ orbital at 162.5 eV (Appendix A), respectively. Compared with the precursor, an obvious red shift of Mo 3d_5/2_ and Mo 3d_3/2_ orbitals was obtained for the MoS_2_ precipitate (below curve in Figure 3D) and the newly appearing peaks at 229 eV (blue line of Mo 3d_5/2_) and 231.1 eV (blue curve of Mo 3d_3/2_) are resulted from the 1T phase of MoS_2_ [5b]. Similarly, the newly emerging peaks at 161.8 eV (blue line of S 2p_3/2_) and 163.1 eV (blue curve of S 2p_1/2_) were observed in the S 2p spectra, which again indicated the presence of 1T phase in the MoS_2_ precipitate (Appendix A) [2,40]. The results of the phase transformation in the as-obtained precipitates through the XPS analysis were consistent with the XRD results. Accordingly, it is concluded that the alteration of MoS_2_ from 2H to 1T phase indeed occurred during the electrolysis process. In addition, the precipitates were mixed with MoS_2_ in both two phases. The XPS spectra of the MoS_2_ QDs in the Mo 3d region which could be divided into four peaks (Figure 3E). Therein, the two intense peaks at 232.8 and 230.0 eV being ascribed to Mo 3d_3/2_ and Mo 3d_5/2_, respectively, should be due to the Mo^4+^ in the MoS_2_ QDs [8,11,41]. The one at 225.5 eV corresponds to the S 2s of MoS_2_ QDs [8,42]. The minor peak locating at 236.0 eV should be due to the Mo^6+^, which corresponds to the slight oxidation of Mo edges of the MoS_2_ upon the exfoliation process [13]. The formation of Mo=O bonds was consistent with the FTIR spectrum (Figure 3B). Meanwhile, the high-resolution S 2p peaks at 168.4 and 163.6 eV represent were ascribed to the sulfide (Figure 3F) [43].

On the basis of the above-mentioned characterizations, the formation mechanism of the MoS_2_ QDs and nanosheets based on the BPE strategy can be proposed as follows: two platinum sheets act as the driving electrode in the BPE system, and the MoS_2_ powder was used as the conductors. When the drive voltage (*E*_total_) of the 5 V was applied on the Pt electrodes, the electrochemical reaction can occur in the solution near both ends of MoS_2_, although there is no direct contact between the Pt electrode and the MoS_2_. The voltage provided by a DC power supply forms an electric field between the cathode and the anode, and the conductor in the electric field (i.e., MoS_2_) produces a polarization potential (*δ*) due to the existence of the electric field. The terminal of the conductor near the driving anode (i.e., the part opposite the anode of the DC power supply in the schematic diagram) is negatively charged and become the cathode of BPE. The conductor near the driving cathode (i.e., the opposite to the cathode of the DC power supply in the schematic diagram) is positively charged and becomes the anode of the BPE. The anodic polarization potential δ^+^ and the cathodic one δ^−^ of the BPE drive the occurrence of electrochemical reactions on the BPE to produce electric current. The total current flowing through the BPE system is divided into two parts: one part flows into the electrolyte solution and one part flows through the suspension of MoS_2_ precursor, which supplied as countless conductors in the BPE systems. It is due to the current flowing through the so-called conductors to cause the electrochemical exfoliation of bulk MoS_2_ to firstly sheets in fewer layers and smaller sizes as well as more defects and phase variation, and then to nanosheets, and finally to the nanodots. Bubbles were observed on the platinum electrodes during electrolysis, indicating the possible evolution of oxygen and hydrogen, which should be favorable for the scissoring of the bulk MoS_2_ to smaller nanosheets and QDs [20]. The as-obtained nanodots could further enhance the formation of the defects and phase transformation of the MoS_2_ and so forth, the longer the electrochemical stripping time of QDs and nanosheets, the higher the yield.

To further elucidate the novelty of this work, comparison of the previous works with the present one has been added in Appendix A [8,13,24,44,45,46]. As indicated, the present work possesses comparable or superiority such as high quantum yield, simple, convenience, environmental-friendliness, etc.

### 3.2. Applications of the As-Obtained MoS_2_ QDs

Cotton fibers were dyed with MoS_2_ QDs aqueous suspension, washed by deionized water, and then dried at 50 °C in the air and finally observed by the inverted fluorescence microscope. As can be seen in Appendix A, the MoS_2_ QDs showed well excitation wavelength-dependent fluorescence for the stained cotton fibers, which are red, blue and green staining under the exposure of green, UV and blue light irradiation (Appendix A). This result confirmed the great application potential of the as-prepared MoS_2_ QDs as valuable fluorochromes in bio-/chem- staining.

Due to the intrisic biocompatiblity of the MoS_2_, the as-prepared MoS_2_ QDs were tested in the bioimaging taking bamboo fiber cells as representatives. Images were collected by laser scanning confocal microscopy (LSCM) after incubation of 5% CO_2_ with MoS_2_ QD for 4 h at 37 °C. More information in detail was presented in the Appendix A. As indicated in Figure 4A, the cells incubated with MoS_2_ QDs exhibited distinct blue PL upon irradiation with a 365 nm excitation light. Due to the small size (<5 nm) of the MoS_2_ QDs, it is easily taken up by cells, enabling efficient bioimaging [47].

The MTT assay was further performed to study the cytotoxicity of the as-obtained MoS_2_ QDs (Figure 4D). The survival rates slightly went down accompanying with the incremental concentrations of the MoS_2_ QDs including 0, 50, 100, 150, 200, 250, and 300 μg mL^−1^, respectively. In the culture medium with the maximum tested MoS_2_ QDs concentration, the cell viability of the cells reached 80%. The good biocompatibility of the as-obtained MoS_2_ QDs to the cells was confirmed by MTT assay.

### 3.3. Application of Precipitate Containing the Nanosheets in EMW Absorption

As reported, MoS_2_ nanomaterials with rich defects and good conductivity possess abundant polarization centers and dielectric relaxation, thus leading to conspicuous dielectric loss, and eventually efficient absorption towards the EMW [9]. Through the analysis of the morphology and structure of the MoS_2_ precipitate as-described above, the precipitate containing the nanosheets with poor crystallinity. This may be caused by a large number of defects before the crystal phase transition occurs. Due to MoS_2_ QDs are formed on the MoS_2_ sheets, cavities and defects on the surface and edge of MoS_2_ precursor were formed, and lots of MoS_2_ nanosheets were created because of the electrolysis process, so the structure of the MoS_2_ precursor was destroyed to a certain extent. Therefore, MoS_2_ precipitate should be a good EMW absorbing material.

To understand the EMW absorption performance of the MoS_2_ precipitate, electromagnetic parameters of MoS_2_ precipitate–wax and MoS_2_ precursor–wax composites with 60 wt % wax loadings were studied in the frequency ranging from 2.0 to 18.0 GHz via the coaxial method. The RL curves of MoS_2_ precipitate–wax and MoS_2_ precursor–wax composites were obtained from the measured electromagnetic parameters at a suggested layer thickness and frequency by the transmit line theory that is indicated based on the following equations (Equations (1) and (2)) [48,49]
(1)$$Z_{\mathrm{in}}=Z_{0}{\left(\mu_{r}/\varepsilon_{r}\right)}^{1/2}\tanh\left[j\left(2\pi fdlc\right){\left(\mu_{r}\varepsilon_{r}\right)}^{1/2}\right],$$
(2)$$\mathrm{RL}=20\log\left|\left(Z_{\mathrm{in}}-Z_{0}\right)/\left(Z_{\mathrm{in}}+Z_{0}\right)\right|,$$
where $\varepsilon_{r}=\varepsilon^{\prime }-j\varepsilon^{\prime \prime }$ and $\mu_{r}=\mu^{\prime }-j\mu^{\prime \prime }$ are the relative complex permittivity and permeability of the absorber, respectively, while $Z_{0}$ is the impedance of free space, c is the velocity of light, f is the frequency of microwaves, $Z_{\mathrm{in}}$ and d are the input impedance and the thickness of the absorber, respectively. Considering the weak magnetic properties of MoS_2_, μ′ and μ″ are taken as 1 and 0, respectively. When RL value is lower than −10 dB, 90% of EMW energy could be absorbed, implying the materials can be used as practical applications.

Figure 5A shows the RL values and effective absorption bandwidths (the frequency range of RL ≤ −10 dB) of MoS_2_ precipitate–wax and MoS_2_ precursor–wax composites at the thicknesses of 3.3 mm. The lowest RL value of MoS_2_ precipitate–wax could reach −54.13 dB at a thickness of 3.3 mm, which was stronger than that of the MoS_2_ precursor–wax composites. The effective absorption bandwidth of 7.12 GHz and 3.28 GHz of MoS_2_ precipitate–wax and MoS_2_ precursor–wax composites were achieved at the thicknesses of 3.3 mm, which means that MoS_2_ precipitate–wax composites exhibit broader absorption bandwidth than MoS_2_ precursor–wax composites. Figure 6B show the 3D plots of RL with different thickness and frequency of MoS_2_ precipitate–wax.

The real parts of complex permittivity (ε′) and imaginary parts (ε″) stand for the storage and the loss capability of electromagnetic energy, respectively. As can be seen from Figure 5C,D, the ε′ and ε″ of the two samples have similar trends with the change of frequency and the values of ε′ and ε″ for MoS_2_ precursor–wax composites are slightly smaller than MoS_2_ precipitate–wax composites. The values ε′ and ε″ are found to descend with the rising of frequency in the 2.0–13.0 GHz. Resulting from the lack of magnetism for the two samples, the real part (μ′) and the imaginary part (μ″) of the complex permeability was independent of the frequency, and remain 1 and 0, respectively.

There are two key factors to evaluate an excellent absorber. One is the impedance matching ($Z=\left|Z_{\mathrm{in}}/Z_{0}\right|$), which requires the equality of the electromagnetic parameters based on Equation (2). When $\left|Z_{in}/Z_{0}\right|$ is close to 1, the EMW could enter into the absorber with greater ease, instead of reflecting into the air, and then the optimal absorption performance can be achieved. The other is the EMW attenuation constant (α) in the interior of the absorber, the larger the value, the more electromagnetic energy is absorbed. The values of α determining the attenuation properties of materials and can be denoted by [50]
(3)$$\alpha =\frac{\sqrt{2}\pi f}{c}\times \sqrt{\left(\mu^{\prime \prime }\varepsilon^{\prime \prime }-\mu^{\prime }\varepsilon^{\prime }\right)+\sqrt{{\left(\mu^{\prime \prime }\varepsilon^{\prime \prime }-\mu^{\prime }\varepsilon^{\prime }\right)}^{2}+{\left(\mu^{\prime }\varepsilon^{\prime \prime }+\mu^{\prime \prime }\varepsilon^{\prime }\right)}^{2}}},$$

Figure 6A gives the frequency dependence of $Z=\left|Z_{\mathrm{in}}/Z_{0}\right|$ and α values for MoS_2_ precipitate–wax composites and MoS_2_ precursor–wax composites in the frequency range between 2.0 and 18.0 GHz. As can be seen from Figure 6A, Z of MoS_2_ precipitate–wax composites are closer to 1. It shows that the Z characteristic of the MoS_2_ precipitate–wax composites is better than MoS_2_ precursor–wax composites, which means that the EMW is easier to enter the absorber of MoS_2_ precipitate–wax composites. Figure 6B exhibits the frequency dependence of the α in the range of 2.0–18.0 GHz. The MoS_2_ precipitate–wax composites have larger α values in the frequency range of 8.2–14.3 GHz, which means compared with MoS_2_ precursor–wax composites more EMW energy is absorbed for MoS_2_ precipitate–wax composites. Therefore, due to the good Z and high α, MoS_2_ precipitate have better electromagnetic absorption properties than that of MoS_2_ precursor.

Based on the Debye theory, ε′ and ε″ can be described as [15]
(4)$$\varepsilon \prime =\varepsilon_{\infty }+\frac{\varepsilon_{s}-\varepsilon_{\infty }}{1+\omega^{2}\tau^{2}},$$
(5)$$\varepsilon \prime \prime =\frac{\varepsilon_{s}-\varepsilon_{\infty }}{1+\omega^{2}\tau^{2}}\omega \tau +\frac{\sigma }{\omega \varepsilon_{0}},$$
where ω is angular frequency, τ is polarization relaxation time, $\varepsilon_{s}$ is static permittivity, $\varepsilon_{\infty }$ is relative dielectric permittivity at high-frequency limit and σ is electrical conductivity. From the Debye theory Equation (4), we can conclude that with increasing of frequency the ε′ value would be decreased, and that is consistent with the results of Figure 5C [51]. Equation (5) shows that ε″ is depended on the polarization and σ.

When the second part of the Equation (4) is not considered, the connection between ε′ and ε″ can be described as
(6)$${\left(\varepsilon \prime -\frac{\varepsilon_{s}+\varepsilon_{\infty }}{2}\right)}^{2}+{\left(\varepsilon \prime \prime \right)}^{2}=\left(\frac{\varepsilon_{s}-\varepsilon_{\infty }}{2}\right),$$

The Equation (6) represents to a circle centered at $\left(\left(\varepsilon_{s}+\varepsilon_{\infty }\right)/2,0\right)$, and every section of the arc form corresponds a Debye relaxation process. Figure 6C indicates the relationship of ε′ between ε″ for MoS_2_ precipitate–wax composites in the frequency range of 2.0–18.0 GHz, and three clear semicircles can be obtained, which indicated that three Debye dipolar relaxation processes benefited to the dielectric loss mechanism for the MoS_2_ precipitate–wax composites [52].

The mechanism of excellent EM wave absorption property for the MoS_2_ precipitate was should be due to the following reasons (Figure 6D). Compared with MoS_2_ precursor material, MoS_2_ precipitate exhibits excellent and enhanced EMW attenuation capability, which results from good Z and high α. The good Z comes from the increasing of the amount of MoS_2_ nanosheets after electrolysis. The high α derives from the enhanced dielectric loss, which mainly caused by the increased polarization relaxation. When EMW propagates into the MoS_2_ precipitate–wax composites, some electrons could hop across the defects and the interface, which could contribute to the dielectric loss [8]. The enhanced dipolar polarization is enhanced by the increased surface point defects. After electrolysis, MoS_2_ QDs are formed and peel off from the MoS_2_ precursor, and plentiful QDs cavities on the surface and edge of MoS_2_ were formed. The point defects could act as polarization centers under an alternating electric field. The interfacial polarization was enhanced, too, which mainly originates from the interface generated by the increased MoS_2_ layers. Also, due to the phase variation from 2H MoS_2_ to 1T MoS_2_ in the process of electrolysis, the existed high conductivity of 1T MoS_2_ in the MoS_2_ precipitate composites has a positive contribution, that is enhanced resistive loss, towards EMW energy attenuation. The efficient EMW performance of the precipitate not only suggested the great application potential of the by-product of the MoS_2_ QDs, but also gave indirect evidence of the defects and phase variation generated during the BPE process.

## 4. Conclusions

In summary, a low cytotoxicity, simple, and nondestructive BPE electrochemical method has been developed to synthesize MoS_2_ QDs with a narrow size distribution and excellent aqueous solubility. Meanwhile, MoS_2_ nanosheets could also be observed in the electrolyzed precipitate, which showed excellent electromagnetic absorption performance. The BPE electrochemical process was not limited with the shape of the MoS_2_ precursor and could lead to the defects and phase variation of the MoS_2_ precursor, finally resulting in the as-prepared QDs and nanosheets. The advantages of this work can be summarized as follows. Firstly, MoS_2_ QDs was produced via simple, one-step top-down method, which is relatively low-cost and environmentally friendly due to the simple electrochemical instrument and low cytotoxicity PBS electrolyte. Secondly, the as-prepared MoS_2_ QDs is hydrosoluble, has low cytotoxicity, and is confirmed to be a promising alternative in PL staining and cell imaging. Furthermore, due to the defects and the phase transformation of the MoS_2_ deduced in the EC process, the as-produced precipitates containing the MoS_2_ nanosheets are highly efficient as an EMW absorber with its strong RL value and broad bandwidth absorption. The BPE design provides a versatile approach to prepare 2D layered nanomaterials in a convenient way. This work would widen the applicability of both the electrochemical methods and the 2D micro-/nano- materials.

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
