# Peer review of "Molybdenum Disulfide Quantum Dots Prepared by Bipolar-Electrode Electrochemical Scissoring"

_nanomaterials, 2019, doi:10.3390/nano9060906_

Reviewer 1 Report

I found the paper interesting and well organized.

To me it can be published as it is.

Author Response

Comments: I found the paper interesting and well organized. To me it can be published as it is.

Response: Thanks for your considerate comments and kind recommendation.

Reviewer 2 Report

This paper describes the preparation of molybdenum disulfide quantum dots prepared by bipolar-electrode electrochemical scissoring and the by-product for ultra-effective electromagnetic wave absorption. It is a good paper showing the utility of a relatively new method for obtaining molybdenum disulfide QDs to be used in various applications. Authors have performed a good characterization study and demonstrated the advantages of the synthesized material toward other reported in the literature. In my opinion, the paper should be published in Sensors after minor revision.

Some specific questions are as follows.

1. The title of the paper is quite complex and should be simplified and shortened. Maybe eliminating the part of the by-product would be the solution

2. A simple application that definitively demonstrated the analytical usefulness of the product obtained is highly recommended.

Author Response

Comments: This paper describes the preparation of molybdenum disulfide quantum dots prepared by bipolar-electrode electrochemical scissoring and the by-product for ultra-effective electromagnetic wave absorption. It is a good paper showing the utility of a relatively new method for obtaining molybdenum disulfide QDs to be used in various applications. Authors have performed a good characterization study and demonstrated the advantages of the synthesized material toward other reported in the literature. In my opinion, the paper should be published in Sensors after minor revision.

Response: Thanks for your good comments and kind recommendation.

Q1: The title of the paper is quite complex and should be simplified and shortened. Maybe eliminating the part of the by-product would be the solution.

A1: Thanks for the reviewer’s good questions. As the reviewer suggested, we simplified the title to "Molybdenum Disulfide Quantum Dots Prepared by Bipolar-Electrode Electrochemical Scissoring". In the revised paper, the yellow background is highlighted at the page 1.

 Q2: A simple application that definitively demonstrated the analytical usefulness of the product obtained is highly recommended.

A2: Thanks for the reviewer’s comments. The applications that definitively demonstrated the analytical usefulness of the products have been indicated in the sections 3.2 and 3.3 pages 8-12), which are summarized as follows:

(1) The prepared MoS2 have excellent photoluminescence properties, low cytotoxicity and good water dispersion, which make them have broad application prospects in photoluminescence dyeing and cell imaging. In order to demonstrate the potential application of the prepared QDs in photoluminescence dyeing and cell imaging, MoS2 were used to dye cotton fibers and cells respectively. The dyed cotton fibers emit red, blue and green fluorescence under green, ultraviolet and blue light, showing good excitation wavelength dependence. In the aspect of cell imaging, bamboo fiber cells were selected as the representative, and the prepared QDs were bio-imaging tested. Images were collected by laser scanning confocal microscopy (LSCM) after incubation of 5% CO2 with MoS2 QDs for 4 h at 37 °C. As indicated in Figure 4A, the cells incubated with MoS2 QDs exhibited distinct blue photoluminescence upon the irradation with 365 nm excitation light. The MTT assay was further performed to study the cytotoxicity of the as-obtained MoS2 QDs (Figure 4D). In the culture medium with the maximum tested MoS2 QDs concentration, the cell viability of the cells reached 80%. The good biocompatibility of the as-obtained MoS2 QDs to the cells was confirmed by MTT assay.

The results of photoluminescence dying and bio-imaging show that the prepared MoS2 QDs have broad application prospects in photoluminescence dyeing and cell imaging.

(2) The prepared by-products containing MoS2 nanosheets can be used as effective electromagnetic wave absorbing materials, and the electromagnetic wave absorbing properties obtained are much better than most reported MoS2-based nanomaterials. The mechanism of good electromagnetic wave absorption performance of by-products should be due to the following reasons. Compared with MoS2 precursor material, MoS2 precipitate exhibits excellent and enhanced EMW attenuation capability, which results from good impedance matching and high attenuation constant. The good impedance matching comes from the increasing of the amount of MoS2 nanosheets after electrolysis. The high attenuation constant derives from the enhanced dielectric loss, which mainly caused by the increased polarization relaxation. When EMW propagates into the MoS2 precipitate-wax composites, some electrons could hop across the defects and the interface, which could contribute to the dielectric loss. The enhanced dipolar polarization is enhanced by the increased surface point defects. After electrolysis, MoS2 QDs are formed and peel off from the MoS2 precursor, and plentiful QDs cavities on the surface and edge of MoS2 were formed. The point defects could act as polarization centers under an alternating electric field. The interfacial polarization was enhanced, too, which mainly origins from the interface generated by the increased MoS2 layers. Also, due to the phase variation from 2H MoS2 to 1T MoS2 in the process of electrolysis, the existed high conductivity of 1T MoS2 in the MoS2 precipitate composites has a positive contribution, that is enhanced resistive loss, towards EMW energy attenuation. The efficient EMW performance of the precipitate not only suggested the great application potential of the by-product of the MoS2 QDs, but also gave indirect evidence of the defects and phase variation generated during the BPE process.

  By measuring the electromagnetic wave absorption parameters (reflection loss, electromagnetic parameters, impedance matching ratio and attenuation constant), it is proved that the by-products of the prepared MoS2 nanosheets can be used as effective electromagnetic wave absorption materials.

Reviewer 3 Report

Ms. Ref. No.: NANOMATERIALS-493320

Title: “Molybdenum Disulfide Quantum Dots Prepared by Bipolar-Electrode Electrochemical Scissoring and the By-Product for Ultra-Effective Electromagnetic Wave 4 Absorption

Authors: Yang Li, Xiaoxia Wang, Mengli Liu, Heng Luo, Lianwen Deng, Lei Huang, Shuang Wei, Congli Zhou and Yuanhong Xu

The manuscript concerns the synthesis of molybdenum disulfide (MoS2) quantum dots (QDs) by bipolar-electrode (BPE) electrochemical method using pure phosphate buffer as the electrolyte and the MoS2 powder as the precursor.. In my opinion this manuscript is interesting, and has got good results, but it needs minor revision before it will be consider for potential publication in Nanomaterials journal. My reservations and comments are given below.

1. Authors and affiliation – affiliation numbers for the personal data of the authors of the publication should appear in accordance with the order of its presentation.  

2. Keywords – Could you refine the keyword “electrochemical”. You meant the “electrochemical method”.

3. In my opinion all abbreviation which appear in the manuscript, should be explain when they appear for the first time.

4. Section 2 – Should be supplemented for all reactants which were used in the presented study. Moreover, in this section should also appear the abbreviation of all materials. For example – page 3, line – 105 – appear abbreviation PBS – Could you explain this?

5. Figure 3 – Please, consider another presentation of individual graphs, because in my opinion, for example, the data presented in Figure 3b are difficult to read.

Author Response

The manuscript concerns the synthesis of molybdenum disulfide (MoS2) quantum dots (QDs) by bipolar-electrode (BPE) electrochemical method using pure phosphate buffer as the electrolyte and the MoS2 powder as the precursor. In my opinion this manuscript is interesting, and has got good results, but it needs minor revision before it will be consider for potential publication in Nanomaterials journal. My reservations and comments are given below.

 Response: Thanks for your good comments and kind recommendation. We have revised the manuscript very carefully based on your valuable comments.

 Q1: Authors and affiliation – affiliation numbers for the personal data of the authors of the publication should appear in accordance with the order of its presentation.

 A1: Thank you for the reviewer’s comments. The corresponding corrections have been added as highlighted with a yellow background in page 1.

 Q2: Keywords – Could You refine the keyword “electrochemical”. You meant the “electrochemical method”.

 A2: Thanks for the reviewer good questions. The term "electrochemistry" in the keywords refers to "electrochemical methods". We replaced the keyword "electrochemistry" in the revised version with "electrochemical method" and highlighted it in the yellow background on page 1 of the paper.

 Q3: In my opinion all abbreviation which appear in the manuscript, should be explain when they appear for the first time.

 A3: Thank for the reviewer's comments. We carefully examined all the abbreviations that appeared in the manuscript and made sure that they were explained clearly when they were firstly appeared. The revised part is highlighted in the revised version with a yellow background accordingly.

Point 4: Section 2 – Should be supplemented for all reactants which were used in the presented study. Moreover, in this section should also appear the abbreviation of all materials. For example – page 3, line – 105 – appear abbreviation PBS – Could You explain this?

Response 4: Thanks for the reviewer’s comments. According to the reviewer's suggestion, all the reactants which were used in the presented study were supplemented in the section 2 and highlighted them in the yellow background at the page 2. Page 3, line 105 – appear abbreviation PBS refers to phosphate buffer solution. The PBS used in this experiment consists of sodium dihydrogen phosphate solution and sodium dihydrogen phosphate solution, which were highlighted in the yellow background in the line 13 of page 1.

Point 5: Figure 3 – Please, consider another presentation of individual graphs, because in my opinion, for example, the data presented in Figure 3b are difficult to read.

Response 5: Thanks for the reviewer’s good suggestions. We increased the number of the annotations in Fig. 3B to make it more clearly shown as highlighted with yellow background on page 6.